## [Decision Letter · Decision Letter 0]

15 Jan 2026

PGENETICS-D-25-01244

Evolutionary turnover of key amino acids explains conservation of function without conservation of sequence in transcriptional activation domains

PLOS Genetics

Dear Dr. Staller,

Thank you for submitting your manuscript to PLOS Genetics. After careful consideration, we feel that it has merit but does not fully meet PLOS Genetics's publication criteria as it currently stands. Therefore, we invite you to submit a revised version of the manuscript that addresses the points raised during the review process.

We look forward to receiving your revised manuscript.

Kind regards,

Michael J. Guertin

Academic Editor

PLOS Genetics

Marnie Blewitt

Section Editor

PLOS Genetics

Aimée Dudley

Editor-in-Chief

PLOS Genetics

Anne Goriely

Editor-in-Chief

PLOS Genetics

**Journal Requirements:**

At this stage, the following Authors/Authors require contributions: Claire LeBlanc, Jordan Stefani, Melvin Soriano, Angelica Lam, Marissa A. Zintel, Sanjana R Kotha, Emily Chase, Giovani Pimentel-Solorio, Aditya Vunnum, Gean Hu, Katherine Flug, Aaron Fultineer, Niklas Hummel, and Max Staller. Please ensure that the full contributions of each author are acknowledged in the "Add/Edit/Remove Authors" section of our submission form.

The list of CRediT author contributions may be found here: https://journals.plos.org/plosgenetics/s/authorship#loc-author-contributions

https://journals.plos.org/plosgenetics/s/submission-guidelines#loc-parts-of-a-submission

5) We have noticed that you have uploaded Supporting Information files, but you have not included a list of legends. Please add a full list of legends for your Supporting Information files after the references list.

State what role the funders took in the study. If the funders had no role in your study, please state: "The funders had no role in study design, data collection and analysis, decision to publish, or preparation of the manuscript.".

**Reviewers' comments:**

Reviewer's Responses to Questions

**Comments to the Authors:**

Reviewer #1: This study examines how transcription factor activation domains (ADs) maintain conserved function despite extreme sequence divergence. Using the yeast transcription factor Gcn4 as a model, the authors analyze 502 homologs spanning ~600 million years of fungal evolution and apply a high-throughput tiling assay in S. cerevisiae to functionally map activation domains across all homologs. This is a technically strong and impressively scaled study that provides a valuable functional dataset. The experimental execution and breadth of evolutionary sampling are clear strengths. However, some aspects of the conceptual framing, statistical choices, and figure interpretation would benefit from clarification.

1. While the manuscript emphasizes that activation function is conserved despite poor sequence conservation, multiple figures (notably Figures 2E and 4) highlight the conserved WxxLF motif around which activation activity is consistently enriched. Although the authors argue that such motifs are neither necessary nor sufficient for activation, this claim is not always clearly reconciled with data showing that mutating conserved motifs reduces activity and that activation peaks localize near these motifs across species. More explicit clarification of how motif contribution differs from necessity or sufficiency across evolutionary contexts would strengthen the central argument.

2. The choice of the top 20% of tiles as the activity threshold in Figure 2 appears pragmatic, but the rationale would benefit from clearer justification. While the Methods note that control sequences define a clearly inactive peak, it is not obvious where these controls fall quantitatively relative to the 20% cutoff. Given the highly skewed activity distribution, it would also be helpful to briefly clarify why a percentile-based threshold was favored over a control- or z-score–based approach. Clarifying how far the chosen threshold lies above control activity, and whether key conclusions are robust to modest changes in this cutoff, would strengthen confidence without requiring reanalysis.

3. The manuscript states that multiple thresholds were tested, but showing how key conclusions change across a small range of thresholds (e.g., 15%, 20%, 25%) would more directly demonstrate robustness.

4. Figure 3J reports that tiles with more published motifs are more active; however, it is unclear how many tiles actually contain multiple motifs. If such tiles are rare, the biological significance of this observation may be limited.

5. In Figure 5E, the comparison between the 50 strongest and 50 weakest central activation domains is not clearly justified. Because activation strength varies continuously, it is unclear whether the observed compositional differences generalize beyond these extremes. A brief rationale for this choice, or clarification that similar trends hold across the full distribution, would strengthen the conclusion without requiring additional analyses.

6. In several comparisons, statistical differences are clear, but the biological magnitude of these effects is less obvious. Briefly reporting or contextualizing effect sizes would help readers assess functional relevance.

7. It would be helpful to briefly restate how variation in tile expression or stability was controlled for or ruled out, particularly when interpreting subtle differences in activity.

8. Figure 1C is difficult to interpret due to the inherent sparsity and indel-rich nature of intrinsically disordered region alignments, and it is not immediately clear how alignment positions relate to known functional regions. A windowed conservation plot along a reference sequence or an occupancy-weighted conservation metric may more clearly convey the intended message. As a minimal alternative, explicitly annotating the locations of the DNA-binding domain (DBD), central activation domain (CAD), and key motifs (e.g., WxxLF) on the current plot would substantially improve interpretability without changing the underlying analysis

Reviewer #2: In this manuscript, LeBlanc and co-workers address the question of whether and how the function of intrinsically disordered transcription activation domains can be conserved without the conservation of sequence. As a model system, they consider the evolution and function of Gcn4 across a diverse group of fungi. Using evolutionary sequence analysis, simulation, and high throughput functional testing, they demonstrate that the function of Gcn4 is generally conserved though there is extensive sequence divergence and highlight specific patterns of sequence turnover within the protein family. They also put forward some intriguing hypotheses, e.g. the prediction of a tight window of activity of the full length Gcn4 homologs. I think the research question is important, as it will push forward our understanding of the function and evolution of activation domains and IDRs more generally, and the approach well-suited to the question. While much of the manuscript is well-written, there were several aspects (some of the logical flow, methods section) for which I have specific suggestions for improvement. The text descriptions of some of the results were also somewhat high-level, making it have to evaluate some of the key claims of the paper. I have detailed these suggestions and questions below.

Major Comments:

- I found that in some sections of the results the logic of some of analysis/experiments was a bit hard to follow, sometimes because the justification of the approach came later in the paragraph or section. I think this might be improved by starting each results subsection with the purpose of the experiment/analysis, instead of the method or finding, and then following with the approach and result. For example, in the section starting on line 114, a topic sentence like “To analyze the patterns of sequence conservation in Gcn4, we ….” would help the reader. The section starting on Line 208 is another place where this structure is missing. I would recommend this structure’s use throughout the results section.

- There were some places where the text description of the results lacked specificity. For example, on line 104, calling out specific numbers from figure 1 to specify the findings about negative selection/drift would be helpful. Adding specific numbers/quantification to the description of the data from Figure 3 is needed to support the claims made in the paragraph starting on line 242. I wasn’t certain how to confidently assess the claim that “these yeast activation domains require a core of aromatic residues…”. Calling out specific results from Figure 2E and 3 will make this claim easier to assess. For example, in Figure 3D, there are many tiles without W…LF motifs that have activity, which seems at odds with the claim as written.

- The Methods section describing the homolog search approach has some information that repeats the results and can be removed, or might be better suited for a supplemental figure legend than the methods (e.g. the discussion of Blastoclaiomycota homologs, the speculation about two vs. one copy of Gcn4 in post-whole-genome duplication species). The information on reciprocal best hit percentage is helpful and might be moved to the results to emphasize the efficacy of the homolog search. In general, the methods section contains commentary that, while helpful, seems beyond the scope of methods (e.g. recommending tiles for CRISPRa studies). As the length of the section makes it harder to search for information, I would recommend streamlining the methods to contain typical methodological information and moving non-methods commentary and discussion to other places in the manuscript (results, discussion, supplementary material), as appropriate.

- I wasn’t sure how to compare the duration of the simulations of neutral sequence drift to 600 MY of divergence. This is somewhat depicted in FIgure 5A, but not discussed in the text.

Minor comments:

- Line 73: maybe give examples of functional elements at the first mention, instead of later in the paragraph?

- Line 89: “The motifs necessary for…” maybe reword this sentence to make it more clear how things can be poorly conserved and experience minimal turnover

- Line 92: at this point in the paper, I didn’t know what “upstream activation domains” meant

- Line 108: A bit of an explanation of the significance of the WxxLF motif and its distance to the DBD here would be helpful.

- Line 150: Does this sentence need to specify that acidic activation domain function is yeast is a reliable measure of function in other organisms, since some domains, e.g. Q-rich, don’t generally function in yeast?

- A brief description of the synthetic TF and GFP reporter on around line 156 would be helpful. A more full description of the GFP reporter in the methods is needed (unless I missed it somewhere?)

- Line 165: Spell out NAD/CAD at their first use in the text.

- Line 183: Re-word the sentence to make clear it is the binding affinity of Med15 to the putative activation domain tiles. In addition, a brief phrase explaining what FINCHES is here would be helpful.

- Line 189: Can the figure legend make more clear what the mutation notation means, e.g. does CAD WxxLF > A mean mutating 5 residues or just W, L, F? I found it in Supp Fig 4, but making it clear here would be helpful.

- Line 240: Spell out SLiM upon first use

- Line 290: Should “only” be in this line?

- Line 304: Is the movement of activation ability enough to conclude recurrent gain and loss (versus insertions/deletions in a linker region between the WxxLF motif and DBD)?

- Line 416: I would only include the p-value of the more suitable statistical test (probably KS, though given the large sample size, either is likely fine)

- Line 963: Citation to InterPro is missing.

Reviewer #3: In this manuscript, the authors carefully investigate the sequence-function relationship in IDRs in transcription factors and how they evolve, using homologs of yeast GCN4 as a model system. How TF activation domains function, especially given their low sequence conservation, is an area of general interest and active study. The authors take advantage of their previously developed high-throughput assay for measuring AD activity to screen for active regions of many homologs of GCN4. The general findings of what features contribute to GCN4 activity are entirely consistent with previously published work. The advance in this study is the careful analysis of how these sequences evolve, and the finding of evolutionary turnover of both individual amino acids and entire activation domains. Overall, I think this is a nice study and a great contribution to the field. There are some aspects of the presentation that I find difficult to interpret, and have suggestions below.

Major comments:

In lines 228-229, 322-333, and figure 5, the authors comment on the conservation of the WxxLF motif. Given that the WxxLF motif was a search criterion for the definition of GCN4 homologs, this conservation is to be expected and isn’t a true result. The authors should note this clearly in these sections of the text.

I believe there is a typo or error in Figure 6B, or this is not showing the turnover of F in this region? What is the salmon highlighted “LF” at the ancestral node right before Kuraishia_capsulata?

The analysis of gain of F before loss of F I find interesting and a nice way to support the idea of turnover. The authors say they found multiple examples of gain before loss. Given the overall prevalence of F in this CAD region, though, how many examples would you expect to see? I assume you also see examples of loss of F where there is no gain that precedes it?

In reference to the discussion section lines 520-538, the authors propose a shift away from thinking about short linear motifs in a permissive context. However, much of the work presented around turnover of single amino acids is in the CAD which contains the well documented and completely conserved WxxLF motif. I understand that dipeptides computationally can predict activity, but in thinking about a molecular model, what do the authors envision these additional F residues are contributing around the conserved motif? Is this not the permissive context around a short motif? It’s well studied how the WxxLF motif is interacting with Med15, and the turnover of D/E residues in this region makes sense with the acidic exposure model. The work presenting turnover of entire upstream ADs in regions that don’t have the conserved WxxLF motif does nicely show that this motif is not necessary for activity. Do the authors suggest these newly arising upstream ADs are also directly interacting with Med15? They present work showing that broadly more active tiles bind Med15 more strongly than less active tiles, using FINCHES. Is that true for these ADs that are outside of the CAD?

Minor comments:

The definitions of the central activation domain, N terminal activation domain, and altCAD are difficult to track. It would be helpful to see that laid out in figure 1, with the domain architecture of S cerevisiae GCN4 with important regions annotated (and locations of the important motifs).

Related, in panel 1D, is the region analyzed that’s called the Central activation domain actually the central activation domain, or a ‘length-matched region around the WxxLF motif” that contains the CAD? Also, is it length matched to the DBD? It would help to have a clearer definition of what exactly is meant by the CAD.

On a similar note, on lines 314 and 315, the authors refer to “138 unique regions around the WxxLF motif”. They also use the term regions in the Figure 5 legend. Regions suggests they are discussion individual positions or groups of positions in the sequence alignment, but I believe they are referring to individual protein sequences in the CAD from different organisms.

The inset in figure 4 needs a scale bar.

On line 440 the authors refer to figure 2I which doesn’t exist; it should be 2G.

Reviewer #4: LeBlanc et al. conducted a comprehensive analysis of the evolution of the sequences and activity of a model IDR, i.e., activation domain in a eukaryotic transcription factor. This work has significant meaning both for our understanding of the principles for IDR evolution, but also more specifically for understanding the evolution of TF activation potential. The latter has been lagging behind studies of the DNA Binding Domain activity due to the more flexible sequence grammar and lack of high-throughput and accurate experimental assay techniques. Therefore, the potential for significant contribution to both evolution and transcription fields of the work is very high. As previous reviewers commented, the scale of the dataset and the multifaceted analyses of the work are both impressive and potentially highly valuable for the community. Below I list my major concerns and suggestions that I hope will help the authors further improve the clarity and focus of their work.

My first major comment concerns clarifying the term "function" and "functionally conserved". What this work measured was the ability of individual 40 aa tiles from >500 Gcn4 homologs to activate a reporter gene. I share a previous reviewer's concern that despite the authors NOT talking about the function of the whole Gcn4, that is naturally what I would assume when I was reading the manuscript. In fact, the authors at times also defaulted to this "whole TF function" when they discuss the conservation of Gcn4's function (since it's the whole TF's function that is related to selection). I actually think that their tile approach is powerful as it allows them to compare homologs of vastly different length and sequence divergence levels. But in the meantime, it is crucial to actively clarify the difference between what they measure and what we usually think of as "function" of a TF, which is the integration of the entire protein and is contingent on the native trans as well as cis environment. Here, we have tiles measured in a single species using a single reporter. This in no way diminishes the impact of their work. I just suggest that the authors scrutinize all instances where "conservation of function" or similar statements are made, and carefully specify the scope of the statement to actively avoid any confusion.

Second, the authors repeatedly mentioned neutral evolution / genetic drift to explain the observed pattern of sequence and activity evolution. In population genetics, neutral theory means evolution at the molecular sequence level is nearly all driven by the fixation of neutral mutations. Hence, the emphasis is on **the lack of or a negligible role of adaptive mutations fixed by positive selection**. However, in most of the places where these terms were used, positive selection wasn't explicitly tested, with the exception of the first part of sequence evolution analyses, where the authors performed an dN/dS analysis. Even there, however, the analysis either didn't test for positive selection or doesn't have the power to detect it (I didn't see where the authors did the branch or site test, which was mentioned in method).

A related issue is that I don't believe one could use a subjective judgement of the distribution of the measured 40 aa tile activity being "tight", even if it is compared to a neutral simulation, to reach the conclusion that the central AD evolved through genetic drift with purifying selection alone. Small changes in activity could well be subject to selection and reflect species adaptation to a unique fitness peak. In enhancer evolution, the eve2 enhancer used to be portrayed as evolving by compensatory evolution under stabilizing selection, leading to the belief that its function remain unchanged over tens of millions of years. However, it was later shown through careful experiments that orthologous enhancers in divergent species generate broadly similar patterns with reproducible and significant differences (Crocket and Erives 2008, PMID: 19043578), suggesting these changes may be locally adaptive. The same could happen to ADs, where the activity is both under stabilizing selection, resulting in the apparent conservation of the central AD and perhaps total activation potential, while different species also experience positive selection in order to adapt to the local optimum for each species' internal and external environment.

Another related issue is how the authors used the term "evolutionary turnover". The authors defined this term as "repeated gain-or-loss of functional elements", which refers to an observed phenomenon. However, on line 397 "One prediction of evolutionary turnover is that ..." it was used as a model to generate predictions, which is confusing. I suggest replacing it (in my understanding) with "compensatory evolution under stabilizing selection". The latter part was implied by suggesting that gain of functional elements permits subsequent loss of other elements in the same sequence, with both steps being selectively neutral.

Lastly, I would like to say that what surprises and interests me the most from this work is not the "conservation of function without conservation of sequence", which has already been demonstrated in enhancers and IDRs (work cited by the authors at the end of Discussion) as a result of flexible grammar. Rather, it is the sheer presence of the central AD across 400 million years of evolution and the fact that its distance to the DBD appears to be highly conserved as well. The latter is surprising to me given the potential for compensatory evolution to result in the "movement" of the AD and the former due to the authors' results suggesting that motifs are not essential (even the WxxLF?).

Experimental question:

** Line 397: Regarding the prediction of the stabilizing selection model, another, perhaps more important, prediction of the compensatory evolution under stabilizing selection is that, to use the example in 6B, having 0 F will fall off of the activity peak (large drop in activity) while having 2F and 1F would have indistinguishable activities. can this be tested?

Other specific comments:

Line 290: "only" feels odd here.

Line 293-295: "These simulations support the idea that the homologs are diversifying under neutral drift while also experiencing negative selection ...". The comparison of simulation to observation does imply purifying selection acting to maintain the activity. But it doesn't provide support for the role of genetic drift (or the lack of positive selection) acting.

Line 315: (W-50 : W+19) please explain this shorthand. Why W+19 here while W+20 in some other figures?

Line 319: “SP motifs” legend says “Black: SP motif” but in the weblogo, it is not easy to identify them as black is used as the color for any S and P residues regardless of whether they are part of the CDK motif. also, it is known that CDK motifs have requirements for the two aa following SP/TP. this should be noted in the text and visualized in the figure.

Line 324-325: "only the WxxLF motif was conserved" isn’t this expected since the regions were selected based on the presence of the WxxLF motif and the MSA was centered on it?

Line 332-333: a previous figure showed that tiles with more motifs on average have higher activity. the trend is especially clear for tiles with more than 3 motifs. the conclusion here is again rested upon the conservation of activation domain function. if that is not true, then gain and loss of motifs can in fact be driving consequential changes in the activity of the AD.

Line 336-337: “D+E conservation matches or exceeds the conservation level of many aromatic residues.” can’t tell from the figure.

Line 342: F108 is indexed on the ScGcn4 sequence? This doesn't match any of the coordinates used in the figure. Maybe add an arrowhead or use a different index? Also, “There are nine more positions where F” there are eleven F residues. the text and the legend both indicate 10 total.

Line 390-391: “we chose 10000 random pairs of tiles and computed the difference between their activities” Are the "random pairs" chosen from the measured tiles or simulated sequence? either way, a more relevant comparison to me is against the distribution of all possible single or double amino acid substitutions, with predicted activity (caveat).

Line 401: “Figure 6B, C, S19” since substitutions happen at the nucleotide level, can you show the nucleotide alignment? Also, in Figure 6B I'm confused by the LF (black font on salmon background) symbol. what does it indicate?

Line 470: “Figure S22” what is "splitTFs" and "PADI"? which one indicates the model (ADHunterLite) trained on the same data as TADA and which on the Gcn4 data?

Line 485-486: example of what I suggested above, i.e., “further supports evolutionary turnover of key F residues in the central activation domain of the Gcn4 homologs.” can be changed to "compensatory evolution under stabilizing selection"

**Have all data underlying the figures and results presented in the manuscript been provided?**

Reviewer #1: Yes

Reviewer #2: Yes

Reviewer #3: Yes

Reviewer #4: Yes

PLOS authors have the option to publish the peer review history of their article (what does this mean? ). If published, this will include your full peer review and any attached files.

**Do you want your identity to be public for this peer review?** For information about this choice, including consent withdrawal, please see our Privacy Policy .

Reviewer #1: No

Reviewer #2: No

Reviewer #3: No

Reviewer #4: No

**Figure resubmission:**
---

## [Decision Letter · Decision Letter 1]

24 Feb 2026

Dear Dr Staller,

We are pleased to inform you that your manuscript entitled "Evolutionary turnover of key amino acids explains conservation of function without conservation of sequence in transcriptional activation domains" has been editorially accepted for publication in PLOS Genetics. Congratulations!

Yours sincerely,

Michael J. Guertin

Academic Editor

PLOS Genetics

Marnie Blewitt

Section Editor

PLOS Genetics

Aimée Dudley

Editor-in-Chief

PLOS Genetics

Anne Goriely

Editor-in-Chief

PLOS Genetics

BlueSky: @plos.bsky.social

Comments from the reviewers (if applicable):

Reviewer's Responses to Questions

**Comments to the Authors:**

Reviewer #1: The authors have addressed my concerns satisfactorily

Reviewer #2: The authors have effectively responded to all my comments and feedback. I feel the manuscript is much improved with the changes made.

Reviewer #3: The authors have addressed all of my comments, so I recommend publication of this article.

**Have all data underlying the figures and results presented in the manuscript been provided?**

Reviewer #1: Yes

Reviewer #2: Yes

Reviewer #3: Yes

PLOS authors have the option to publish the peer review history of their article (what does this mean? ). If published, this will include your full peer review and any attached files.

**Do you want your identity to be public for this peer review?** For information about this choice, including consent withdrawal, please see our Privacy Policy .

Reviewer #1: No

Reviewer #2: No

Reviewer #3: No

**Data Deposition**

http://datadryad.org/submit?journalID=pgenetics&manu=PGENETICS-D-25-01244R1

**Press Queries**

---

## [Editor Report · Acceptance letter]

PGENETICS-D-25-01244R1

Evolutionary turnover of key amino acids explains conservation of function without conservation of sequence in transcriptional activation domains

Dear Dr Staller,

We are pleased to inform you that your manuscript entitled "Evolutionary turnover of key amino acids explains conservation of function without conservation of sequence in transcriptional activation domains" has been formally accepted for publication in PLOS Genetics! Your manuscript is now with our production department and you will be notified of the publication date in due course.

With kind regards,

Anita Estes

PLOS Genetics

On behalf of:
